# RNA Polymerase and Transcription Mechanisms: The Forefront of Physicochemical Studies of Chemical Reactions

**DOI:** 10.3390/biom11010032

**Published:** 2020-12-29

**Authors:** Nobuo Shimamoto, Masahiko Imashimizu

**Affiliations:** 1National Institute of Genetics Mishima, Shizuoka-ken 411-8540, Japan; 2Cellular and Molecular Biotechnology Research Institute, National Institute of Advanced Industrial Science and Technology, Tokyo 135-0064, Japan; m.imashimizu@aist.go.jp

**Keywords:** transcriptional regulation, reaction theory, prediction of promoters, one-dimensional diffusion, rate equation, detailed balance, antenna effect, physicochemical techniques

## Abstract

The study of transcription and its regulation is an interdisciplinary field that is closely connected with genetics, structural biology, and reaction theory. Among these, although less attention has been paid to reaction theory, it is becoming increasingly useful for research on transcription. Rate equations are commonly used to describe reactions involved in transcription, but they tend to be used unaware of the timescales of relevant physical processes. In this review, we discuss the limitation of rate equation for describing three-dimensional diffusion and one-dimensional diffusion along DNA. We then introduce the chemical ratchet mechanism recently proposed for explaining the antenna effect, an enhancement of the binding affinity to a specific site on longer DNA, which deviates from a thermodynamic rule. We show that chemical ratchet cannot be described with a single set of rate equations but alternative sets of rate equations that temporally switch no faster than the binding reaction.

## 1. Introduction

The intellectual horizon of biological studies started within the fields of replication, transcription, or translation several decades ago. From a historical point of view, transcription has developed uniquely alongside physicochemical and biophysical techniques due to the early establishment of a fully active reconstitution system with purified components among the three above mentioned fields [1]. The close relationship with chemistry and physics has had both positive and negative effects on these biological fields. Most physicochemical techniques have been applied in the study of transcription to open their application in biology. In contrast, their careless use due to a lack of understanding their limitations has tended to result in misleading interpretations. The recent advances and degeneracy of the techniques are summarized by Buckle et al. in this special issue.

Another historical event in the study of transcription was the early discovery of an initiation factor, the *E. coli* sigma-70 subunit. Sigma-70 is able to activate the synthesis of most native transcripts, contributing to reconstitution with purified factors and the identification of promoters. The construction of a fully active reconstitution system was initially expected to deductively interpret physiological phenomena in chemical and physical terms, thereby unifying the basic principles of biology, chemistry, and physics. Contrary to this expectation, mechanistic studies have been inductive rather than deductive.

Since DNA-dependent RNA polymerase (RNAP) has an enzyme with template DNA as a cofactor, genetic information has provided a high number of correlations between DNA sequences and the functions of RNAP. Thus, mechanistic biochemical studies have been combined with genetics to conduct inductive studies using genetic information. This tendency is typically found in the prediction of transcriptional promoters. In the sigma-70 promoter of *E. coli*, several functional motifs have been proposed, a −10 motif and a −35 motif, for example [2,3,4,5,6,7]. Imashimizu et al. combined statistical mechanics with high-throughput sequencing to characterize abortive transcription and pausing during transcription initiation [8].

Although genetics has been a powerful tool in mechanistic studies, it has reached a saturation point in the field of transcriptional regulation. Despite this, there are movements beyond the conventional limits of genetics, such as the polymorphic features of a single gene product, as discussed by Chatterji and colleagues in this issue. Thus, the structure, especially the dynamic structure, is critical in this case, indicating the limitations of X-ray structural analysis. Attempts have been to overcome this limitation with molecular dynamics, as exemplified by Génin and Weinziel in this issue.

DNA in the study of transcription is not limited to sequence information. Transcription factors and RNAP, as well as its complexes, are able to diffuse along DNA. For example, this diffusion is involved in promoter search by RNAP, in Brownian ratchet as an elongation complex, and in backtracking of initiation and elongation complexes. This diffusion, generically known as “one-dimensional diffusion”, may be a distinct mechanism of transcriptional regulation. However, it is difficult to model the one-dimensional space for the diffusion to reflect the real polymeric structure of DNA, as discussed later.

As in other biological processes, transcription is composed of chemical reactions. Notably, there are several basic requirements for a chemical reaction to be described with a rate equation [9]. As will be discussed below in detail, a bimolecular association process mediated by diffusion can be described with rate equations in some cases. However, the transfer itself is beyond the description with rate equations. Furthermore, consideration of the timescales of reactions is critical since timescale matching is essential for the functions of regulators to be expressed, as will be discussed later. Until recently, the mechanism of the chemical ratchet had been overlooked because of the lack of these notions. However, understanding the foundation of scientific tools is essential to promote soundness and clarity in science.

## 2. Implicit Assumption of Rate Equations and the Danger of Applying Them to Proteins and DNA

In chemistry, the rate equation is the most representative analytical method and has been developed over several centuries. The reaction rate or reaction velocity is described as Equations (1) and (2) for a unimolecular reaction or bimolecular reaction, respectively, using the time-independent constant k. However, these equations are not universal truth, and they are based on several assumptions as summarized in Ref [9]. The most essential assumption is the homogeneity of the reactant in terms of the reaction. Equations (1) and (2) assume that all reactant molecules share the same probability of being converted into the product. If the reactant is inhomogeneous, the unreacted reactant would be enriched over time, contradicting the time independence of the rate constant. However, this essential requirement is not emphasized in many kinetic textbooks.
(1)$$\left(reaction\;rate\right)\;=\;k\left[reactant\right]$$
(2)$$\mathrm{or}\;\left(reaction\;rate\right)\;=\;k\left[reactant_{1}\right]\left[reactant_{2}\right]$$

Let us consider the case of the bimolecular binding reaction A + B→C. Among the A reactant molecules, different molecules have different distances from their nearest B molecules with different velocities, namely, different initial conditions and/or different molecular histories (where the molecules are positioned, how large and in which direction were their momenta in the past). The possibility of the reaction of an A molecule, in principle, depends on its molecular history. Obviously, the shorter the distance between A and B, the larger the possibility of producing C. The locations and velocities of these molecules are randomized due to their rapid and numerous collisions with solvent molecules. The collision with other A molecules and the non-productive collisions with B molecules also contribute to the randomization. Even when some of the reactant molecules have large velocities, namely higher local temperatures, all the reactants will have a common temperature over time owing to their collisions before the reaction under consideration. This time-consuming convergence is conventionally termed “thermal equilibrium”. To avoid the ambiguity, we will use “molecular shuffling” instead of “thermal equilibrium” hereafter, since thermal equilibrium is also used between the two macroscopic states in a chemical reaction.

When the molecular shuffling is sufficient, the overall average of the reaction possibilities converges to a unique constant, 1/k or 1/$k\left[B\right]$, which is independent of the molecular histories and time. In other words, the rate constant can be defined only when the “shuffling” is sufficient to homogenize the reactant molecules (see a stricter statistical discussion in [9]). That is, reactant molecules must be confined to the reactant state until shuffling becomes sufficient. Therefore, we can define a reactant only when it is in a state of being completely surrounded by potential barriers that are much larger than the average energy of thermal fluctuation $\left(\frac{1}{2}k_{B}T\right)$. The requirement of enough molecular shuffling excludes all thermal diffusion from the description with rate equations. Notably, the molecular shuffling of macromolecules is slower than that of small molecules because of their higher molecular weights and could be even slower than the reaction under consideration, as will be discussed later.

## 3. Chemical Reactions beyond Rate Equations

The most famous extrapolation beyond the essential assumption of homogeneity is the rate constant for a diffusion-controlled bimolecular reaction, where every collision between reactant molecules results in the formation of complexes. The rate constant is given by the Debye–Smoluchowski equation:(3)$$k_{diff}=\frac{4\pi r}{1000}\left(D_{protein}+D_{DNA}\right)N_{A}$$

In this form, *r* is the encounter distance between the protein and DNA, and $D_{protein}$ and $D_{DNA}$ are their diffusion coefficients. $N_{A}$ denotes Avogadro’s number, and the number 1000 must be added when the concentration and length are expressed in molarity and cm, respectively.

The elementary process of the encounter is the thermal diffusion of the reactant molecules. In the case of diffusion-controlled binding, molecular shuffling is largely limited because most collisions between A and B produce C. Thus, a close pair of A and B molecules reacts more frequently than a remote pair, and thus the reactants are inhomogeneous. Therefore, $k_{diff}$ is calculated using a theoretically inconsistent framework, as mentioned previously [10].

In contrast, if the bimolecular reaction is not diffusion-controlled (i.e., most collision events between A and B result in their dissociation, and only a small fraction of the collisions are productive), shuffling must be mediated by non-productive collisions to rationalize Equation (2). It may be possible to use the value of $k_{diff}$ in Equation (3) as the upper limit of the rate constant of bimolecular association. However, $k_{diff}$ cannot be used in a rate equation as a rate constant.

## 4. Difficulties in Segment Models of One-Dimensional Diffusion

For the analysis of the one-dimensional diffusion of proteins along DNA, the segment model of DNA has contributed to the understanding of the various aspects of the diffusion. In this model, linear DNA is divided into segments, and one-dimensional diffusion is defined as the transfer between the segments with a rate constant (Figure 1a). The size of the segment is usually set to the protein size without any overlaps. When the diffusion is driven by thermal fluctuation, it cannot be described with rate equations, as already mentioned above. When the diffusion requires significant activation energy, there could be sufficient molecular shuffling, but the diffusion becomes more difficult. Moreover, it is difficult to rationalize that a potential barrier exists at every segment ends but not its inside (Figure 1a).

The size of the protein-binding site is different from the distance between contiguous protein-binding sites (Figure 1a,b). The distance between the sites is one base pair irrespective of the size of the protein because a protein-binding site is defined at every structural repeat of the polymer (i.e., one base pair). In contrast, the site size is usually equal to the size of the protein on the DNA, typically 15–40 base pairs, unless the protein forms a ring or helical polymer with DNA at its axis, such as LecA [11]. Therefore, at low protein concentrations where the diffusion along the exposed region of DNA takes place with few collisions, there are much more empty protein-binding sites on DNA than those the segment model supposes. These binding sites are at least accessible from bulk. When the effect of nonspecific sites is discussed with a segment model, this entropic effect should be taken into account.

The competitive nature of nonspecific binding to DNA is required to evaluate the number of nonspecific complexes. Furthermore, the binding at a site should sterically block additional neighboring sites on both sides. The length of the blocked sites is determined by the protein size as well as the shape of the protein complexed with the helically arranged binding sites on DNA. Since this characteristic mode of competition and the high density of nonspecific sites are based on the structure of double-stranded DNA, it is not easy to incorporate these aspects of one-dimensional diffusion in the segment model.

In a single-molecule analysis, protein concentrations are sometimes increased to the level where several or more protein molecules are complexed on a single DNA chain to obtain enough samples for statistical averaging. The segment model is difficult to rationalize at such high protein concentrations. At present, one-dimensional diffusion can be analyzed strictly by a diffusion equation on a continuum DNA only at low protein concentrations where the collision between two protein molecules on the same DNA molecule is ignored. The construction of a theoretical framework based on overlapping binding sites, as shown in Figure 1b, has been a significant challenge in biology.

## 5. Timescale: Index Showing How Fast a Reaction Becomes Stationary

The timescale is defined as the time required for a converging reaction or phenomenon to become stationary. In the case of the exponential decay of $e^{-t/\tau }$, the timescale is defined by the time constant τ. Otherwise, it is the time required for the deviation from the stationary value to reduce by half or $e^{-1}$. In kinetic analysis, the reactions with much faster (or shorter) timescales are considered to be equilibrated, and those with much slower (or longer) timescales are assumed to be stopped. Reactions with similar timescales are called coupled and must be treated as a dynamic phenomenon. This coarse graining of time is at the heart of the temporal analysis.

Notably, the timescale defined above is specified in both forward and backward reactions. If the reaction is $A\begin{array}{c} k^{+}\\ ⇄\\ k^{-} \end{array}B$ or $A+B\begin{array}{c} k^{+}\\ ⇄\\ k^{-} \end{array}C$, then the timescale is $\tau \;=\;{\left(k^{+}+k^{-}\right)}^{-1}$ or $\tau \;=\;{\left(k^{+}\bar{\left[A\right]}+k^{-}\right)}^{-1}$, respectively, where $k^{+}$ is the association rate constant, $k^{-}$ is the dissociation rate constant, and $\bar{\left[A\right]}$ is the temporal average of A that exists in excess over the other reactant B.

We can rephrase the necessary condition for the use of the rate equation as that the timescale of the molecular shuffling that is much faster (shorter) than that of the reaction under consideration. This is usually satisfied for the chemical reaction of small molecules. However, the conformational changes for macromolecules can be slower than the reactions under consideration, the timescales of which are minutes, hours, and even years [12,13,14].

## 6. Transcriptional Regulation by Timescale Matching and Mismatching

When two or more pathways exist in tandem in a stepwise process, the timescale of the whole process is their sum. The transcription process contains many stepwise reactions, and its regulation is critically determined by their summed timescale. Let us consider one of the most typical types of transcriptional regulation in *E. coli*, the competitive binding between a repressor and RNAP at a promoter overlapping with an operator. The step is followed by the formation of ternary complexes of RNAP, DNA, and transcripts. It is widely believed that transcriptional inhibition is determined only by two factors: the concentration of the repressor and its affinity for the operator. This is true only when the timescale of the repressor binding matches the timescale of the production of the mature transcript, as follows.

Suppose that there is a repressor (I), RNAP, and promoter DNA, and that both I and RNAP exist in excess over the promoter in a cell. The binding reactions of I and RNAP to the DNA are assumed to be more rapid than the formation of the first phosphodiester bond, which is a likely assumption in a cell. The promoter is then fractionated into a free promoter, an inactive promoter complexed with repressor, and an active promoter bound by RNAP in the open complex by the ratio of $1\;:\;\left[I\right]/K_{I}\;:\;\left[\mathrm{RNAP}\right]/K_{R}$, respectively, where $K_{I}$ is the dissociation constant of the repressor–operator complex, and $K_{R}$ is that of the RNAP–promoter open complex. During initiation, the total amount of the promoter DNA available in this equilibrium is reduced by the formation of the initiation ternary complex before promoter clearance, promoter + RNAP + transcript. Each of the first four phosphodiester bonds is formed in 30–100 ms on the strongest T7A1 promoter during initiation [15]. The inhibitor decelerates the formation of the ternary complex by reducing the fraction of $\left[\mathrm{RNAP}\right]/K_{R}$, typically by less than a second. When the concentration of I is close to or higher than $K_{I}$, the deceleration of the formation is significant in the production of the ternary complex (Figure 2a; unless $\mathrm{RNAP}\gg K_{R}$). However, when the timescale of the ternary complex formation delayed by the repressor binding is still faster than the timescale of the production of the mature transcript (typically in the order of minutes), the repressor only slightly inhibits transcription due to the mismatching of the timescales (Figure 2b).

To realize the inhibition by the repressor, its binding reaction must be as slow as the production of mature transcripts. A realistic way is to introduce a time-consuming step between covalent bond formation/cleavage in the binding is to use a regulator such as AraC. The phosphorylation of the two-component system can function similarly, in addition to its main function in signal transduction. Furthermore, when a corepressor-regulator complex has two or more conformations with different affinities for its operator, this repressor-operator binding can be delayed by the conformational changes. Abortive transcription can assume a similar function by forming an inactive ternary complex, which may be one of its biological functions. With timescale matching, the repressor introduces a longer delay in the production of a mature transcript with a longer time lag (Figure 2c).

Furthermore, if there is a time-consuming elongation pause or RNA processing, timescale matching can again be hampered by further deceleration in the production of mature transcripts (Figure 2d). If such a pause introduces a long time lag, sensitivity of the transcription to the repressor will be reduced. Since these changes by timescale matching and mismatching can work as the fine tuning of gene expression, it will be interesting to classify the known regulatory mechanisms and discover new cross-talk in transcription from this perspective.

In a cell, transcripts are not necessarily stable, and some are degraded rapidly. The coexistence of synthesis and degradation is called the “futile cycle” or, more positively, the “push–pull” mechanism. Although there is a 2000-fold difference in the timescale between the examples shown in Figure 2a,d, RNA degradation alters the timescale of RNA accumulation faster (shorter), making timescale matching easier than the process, as illustrated in Figure 2c. This could be another function of RNA degradation.

## 7. Chemical Ratchet

For a long time, the first step in kinetic analysis was to define the homogeneous reactants. However, one must first determine whether the reaction contains a chemical ratchet, which was proposed recently for protein–DNA binding [16]. This new concept is an extension of the ratchet mechanism in physics, whose driving force was originally an external energy source [17], but the force has been extended to internal ones [18,19], which is the case for chemical ratchet also.

Figure 3a shows a kinetic scheme for the simplest unimolecular reaction, the inter-conversion between A and B (Figure 3a, “Mechanism a” hereafter). A and B are the two states of the reaction components. These states can be extended to more general states. If the reaction is a binding one, A is composed of two dissociated reactants and B is their complex. At equilibrium, the thermodynamic rule of the detailed balance should hold.

When B exists in two conformations with different reactivities, B_1_ and B_2_, the kinetic scheme becomes Mechanism b (Figure 3b) with a conformational change of the $k_{c}^{\pm }$ step. This conformation can change in the direct pathway of $k_{c}^{\pm }$ and/or in the stepwise $k_{1}^{\pm }+k_{2}^{\pm }$ pathway via A. The forward reaction of $k_{1}^{\pm }$ stochastically coexists with that of $k_{2}^{\pm }$ at all times. At equilibrium, the detailed balance also holds.

A chemical ratchet is shown in panel c and is composed of alternative reaction pathways that switch spontaneously. The kinetic scheme for the chemical ratchet resembles that of Mechanism b. The key difference exists at the molecular or microscopic level. The two pathways of $k_{1}^{\pm }$ and $k_{2}^{\pm }$ are alternatives in the chemical ratchet, while they coexist in Mechanism b. In the chemical ratchet, the pathway is $k_{1}^{\pm }$ for $t_{1}$ and then switches to $k_{2}^{\pm }$ for $t_{2}$ before again switching back to $k_{1}^{\pm }$ for $t_{3}$, and so on. In the pathway $k_{1}^{\pm }$ or $k_{2}^{\pm }$, B exists in the form of either B_1_ or B_2_, respectively, which never coexists at molecular level in the chemical ratchet. In other words, in Mechanism b, the conversion between B_1_ and B_2_ is possible at all times, while these conversion pathways do not exist in the chemical ratchet. In this mechanism, the interconversion is only possible at the time of switching. Thus, the potential mean force switches alternatively, as shown in Figure 3c.

With regards to the timescale of the switching, the average of $\left(t_{odd}+t_{even}\right)$, is coupled with or slower (larger) than the timescales of the reactions ($k_{1}^{+}+k_{1}^{-}{)}^{-1}\;\mathrm{and}\;{\left(k_{2}^{+}+k_{2}^{-}\right)}^{-1}$; otherwise, Mechanisms b and c would become kinetically identical. In other words, the internal degree(s) of freedom corresponding to the switch is the slowest, or close to the slowest, among all degrees of freedom. Because the detailed balance holds at the equilibrium of the reaction with the slowest timescale [9], and because the timescales of the $k_{1}^{\pm }$ and $k_{2}^{\pm }$ reaction are not the slowest, a detailed balance in the reactions of $k_{1}^{\pm }$ or $k_{2}^{\pm }$ is not guaranteed. Deviation from the detailed balance by switching the potential mean force has already been reported [18,19].

Moreover, in a chemical ratchet, the reaction of $k_{1}^{\pm }$ or $k_{2}^{\pm }$ is not equilibrated after switching but oscillates between the two imaginary equilibria of the reactions. As $\;k_{1}^{+}>k_{1}^{-}$ in the example shown in Figure 3c, the net flow from A to B_1_ is going to equilibrate A and B_1_. Since $k_{2}^{+}<k_{2}^{-}$ (as shown by the potential $k_{2}^{\pm }$), an opposite net flow from B_2_ to A is going to equilibrate B_2_ and A. Therefore, at a microscopic level, periods of non-equilibrium exist in the chemical ratchet.

The induced microscopic non-equilibrium is an oscillation, and the time average of many cycles of oscillation converges to a constant as the averaging period becomes longer. Similarly, because the phases of many microscopic oscillations are independent and random, the ensemble average also converges to a constant as the size of the ensemble becomes larger. Therefore, in chemical ratchets, microscopic non-equilibrium states exist in a macroscopic stationary state. Although this stationary state is the time-independent state to which the system converges, the term “equilibrium” is reserved here to describe this state because it contains non-equilibrium.

## 8. A Possible Molecular Example and the Detection of Chemical Ratchet

A possible example of the molecular model for B_1_ and B_2_ can help describe the reality of a chemical ratchet. The DNA B-helix has rigidity with a persistent length of ca. 150 bp. The B-helix in the protein complex is made even more rigid due to the interactions maintaining its specific complex. This enhanced rigidity of DNA can produce a chemical ratchet (Figure 4a).

The reaction $k_{1}^{\pm }$ with Potential 1 involves the binding of proteins to a straight specific site mediated by one-dimensional diffusion through tracking the DNA glove, a mode of one-dimensional diffusion called “sliding”. The longer the DNA, the larger the rate constants $k_{1}^{+}$ and $k_{1}^{-}$. Thus, sliding out from the specific site is involved in the primary dissociation pathway on Potential 1. The potential mean force switches to Potential 2 when the rigid specific site in the stable complex B_1_ is bent by high-energy thermal fluctuations to form an unstable complex B_2_. This bending occurs infrequently because it requires a large amount of energy, and thus the values of $t_{even}$ are long and make the timescale of the switching long irrespective of the values of $t_{odd}$. Thus, the timescale of the switching can be longer than or close to that of the binding reactions, satisfying the requirements of chemical ratchet. In this example, Complex B_2_ is distributed on the high-energy slope of the potential mean force according to its bend angle, as illustrated in Figure 4a. Because the groove structure in B_2_ is distorted by the bend, its dissociation through sliding out is inhibited. Therefore, B_2_ is destined to dissociate directly into A independently of the DNA length. In this way, the potential mean forces of 1 and 2 alternate. Notably, because B_2_ is not confined by potential barriers, there is little molecular shuffling before its dissociation, indicating that the rate constant $k_{2}^{-}$ cannot be defined as mentioned in Section 2. Although the potential around B_2_ is not the same as that shown for B_2_ in Figure 3c, this example also satisfies the requirement for the chemical ratchet.

In Potential 1, both the $k_{1}^{+}$ and $k_{1}^{-}$ reactions are increased for longer DNA, while their ratio, the affinity, are independent of DNA length. If switching occurs, the major association occurs through Potential 1, which is length-enhanced, while dissociation significantly proceeds through Potential 2, which is length-independent. Consequently, the affinity of the protein for a specific site is increased by its DNA length.

The theoretical framework of chemical ratchet was proposed by Toda based on the experimental results and interpretation by Kinebuchi and Shimamoto with mathematical support by Nara et al. [16]. In the experiment of *E. coli* TrpR binding to *trpO*, the apparent dissociation equilibrium constant changed by 10,000-fold according to the length of the DNA harboring *trpO*, which was denoted as the antenna effect [10,20]. The scenario shown above is one of the possible candidates for molecular models of chemical ratchets. As shown in Figure 4b, the observed values of the dissociation equilibrium constant, the inverse of the affinity, are well fitted by the solution of the differential equation composed of the diffusion equation and the rate equation (blue curve in Figure 4b) [16]. For short DNA, where sliding is assumed to be equilibrated before dissociation from the DNA, the concentration of the complex can be dynamically determined from kinetic equations. The apparent dissociation constant is given as the temporal average over a period longer than the timescale of the switching (red curve in Figure 4b). The fitting with a consensus site size of 18 bp for *trpO* is satisfactory [21].

At present, there is also another mechanism exerting the antenna effect: the looping mechanism. If a single protein molecule has two binding sites for DNA, the complex formed at the first binding can be further stabilized by the second binding. As the DNA becomes longer, more possibilities exist for second binding with the same DNA molecule by forming a DNA loop. This has been proven for several proteins with two DNA binding sites, including homodimers of proteins with single binding sites, [22,23,24]. However, there are no proteins with a single DNA binding site, such as TrpR, to be shown in looping mechanism. Moreover, we experimentally denied this possibility for TrpR by using a DNA connection that retains DNA looping but blocks one-dimensional diffusion [21]. This study also provided evidence for the existence of antenna effect caused by one-dimensional diffusion in vivo. All these lines of evidence are, thus far, consistent with the chemical ratchet of TrpR binding to *trpO*.

## 9. Biological Significance and Indications of Chemical Ratchet

The specificity of a protein is defined as the ratio of its affinity for the specific site to that for a nonspecific site. The parameter is a critical factor for determining the minimum level of a protein in cell. There is a 10-fold discrepancy between the values of the specificities of TrpR-*trpO* binding in vitro [25] and those expected from the TrpR protein levels in vivo [26], which was originally reduced to the possible different conditions. Notably, if the antenna effect measured under similar conditions in vitro is considered, this discrepancy will disappear [21]. Antenna effect can decrease the minimum level of a protein through chemical ratchet mechanism.

A new type of cross talk can be predicted from the antenna effect caused by the chemical ratchet in the presence of one-dimensional diffusion. When a regulatory protein has an antenna effect due to one-dimensional diffusion along DNA, the binding of a foreign protein near the operator site of the regulatory protein will decrease the affinity of the regulatory protein for the operator by hindering its one-dimensional diffusion [21]. This cross talk at a distance can be universal. The binding of a repressor near a promoter could block the diffusion of RNAP onto the promoter even when there is no direct steric hindrance between their bindings.

How can we decide whether this mechanism is a chemical ratchet? The most direct method is to detect the imbalances of the microscopic flow in a macroscopically stationary state by a single-molecule experiment. However, this is usually too tedious for the determination of a reaction scheme. Furthermore, the quantitative analysis of single-molecule experiments is always exposed to the danger of artifacts caused by surface effects and fixing methods; thus, a good control experiment is essential.

At present, the biological significance of microscopic flow other than the antenna effect remains unclear. Forty years ago, the dimerization of yeast enolase was combined with a slow conformational change to suppose a mechanism similar to chemical ratchet [27]. The mechanism was then attached as a violation of detailed balance [28,29], which is now answered by chemical ratchet.

## 10. Future Prospect

We now have the chance to find a new mechanism and new cross talk in the correct framework of the reaction theory of DNA-binding proteins. Keeping in mind the timescales of the reaction under consideration and relevant physical and chemical processes, one can select a suitable theoretical framework or kinetic schemes in the search of new mechanisms. Here, we limit our discussion to transcription, but the discussion is easily extended to other biological fields. Chemical ratchet provides a unique example of non-equilibrium but stationary state where microscopic fluctuations yield physiological consequence. The maturation of transcription studies in this direction facilitates challenging feedback from biology to chemistry with the depth of both physics and biology.

## Figures and Tables

**Figure 1 biomolecules-11-00032-f001:**
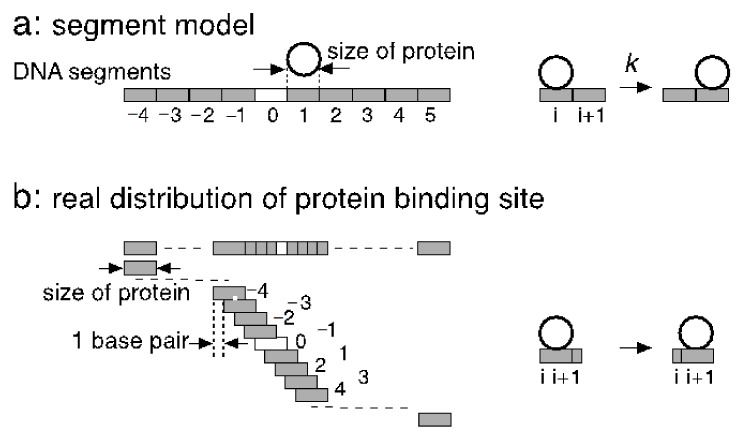
Segment model for one-dimensional diffusion and its difficulties. DNA is considered to be a sequence of binding sites. A specific site is illustrated as an open box, and a nonspecific site as a gray box. (**a**) In the segment model, the binding sites do not overlap with each other, and the site size is usually assumed to be the size of the protein (open circle). Diffusion is replaced by a transfer reaction in the next box with a rate constant *k*. (**b**) More real definitions and distributions of protein binding sites on DNA. A protein binding site exists at every base pair regardless of the protein size. This real distribution reflects the repetitive structures of nucleotides and is independent of the site size. Therefore, DNA is a sequence of overlapping boxes, and one-dimensional diffusion is a movement of the gravity center of the protein along the DNA that cannot be described with a rate constant because of the absence of molecular shuffling.

**Figure 2 biomolecules-11-00032-f002:**
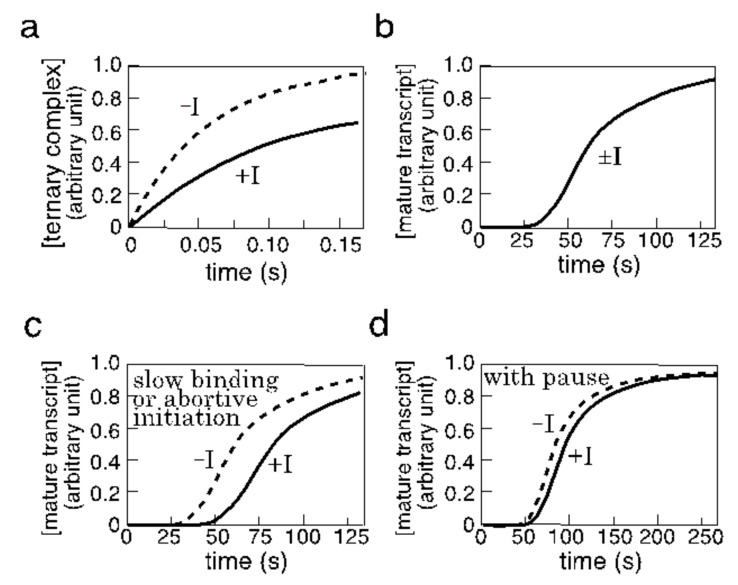
Competitive repressor binding and transcription inhibition. The time courses of ternary complex formation (**a**) and the production of the mature transcript (**b**–**d**) are shown in the case where the repressor (I) and RNAP (R) competitively bind to the promoter–operator overlapping domain of the DNA. The binding reactions are supposed to be rapid, if compared with subsequent reactions. These time courses are generally dependent on many experimental parameters, as well as RNA degradation, and are thus presented as example images. The scale of the coordinates, time, also symbolically indicates the differences between the timescales. (**a**) The addition of a repressor can significantly delay the formation of the ternary complex. The timescales are decreased by half in this example. (**b**) When the timescale of the formation mismatches that of the production of the mature product, the repressor cannot inhibit transcription. The two curves with or without the repressor almost overlap each other. (**c**) When slow binding or abortive initiation makes the timescale of the formation slower (longer), then the inhibition of ternary complex formation is reflected in the production of the mature transcript. (**d**) If an elongation pause is introduced, the mismatch is again recovered, and the inhibition is reduced. RNA processing can exert a similar effect to the pause.

**Figure 3 biomolecules-11-00032-f003:**
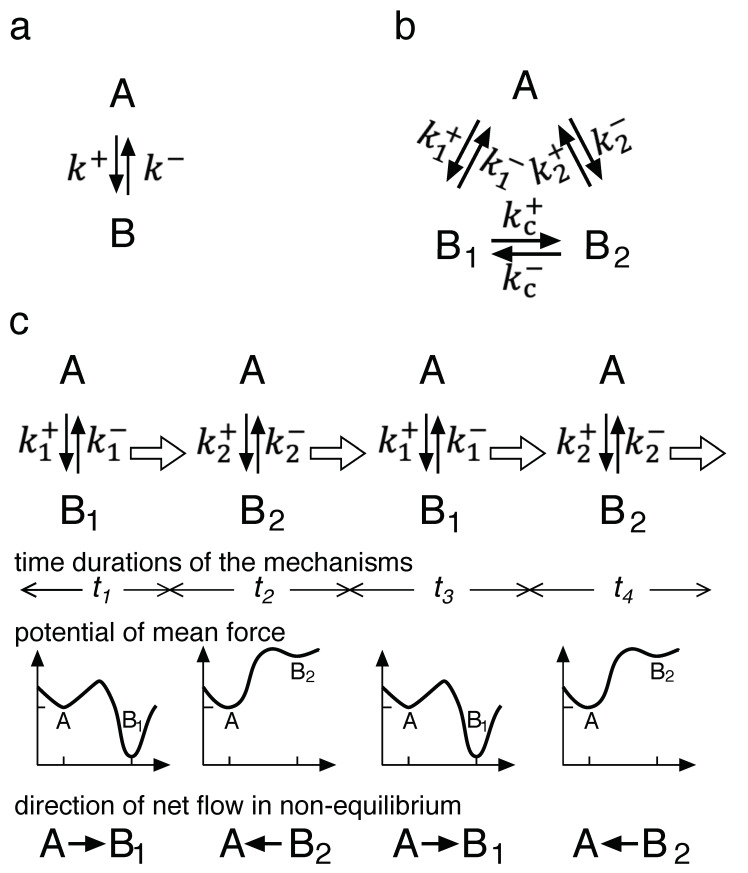
Chemical ratchet. (**a**) The simplest kinetic scheme composed of two states, A and B, which can be described with the rate constants ($k^{\pm }$). (**b**) The kinetic scheme when the B state is divided due to inhomogeneity in the reactant with two different activities. (**c**) The scheme of a chemical ratchet with two alternating mechanisms of reaction $k_{1}^{\pm }$ and reaction $k_{2}^{\pm }$. The switching is indicated with open arrows, and its timescale must not be faster (shorter) than that of the conversion between A and B_i_. Below the scheme, the time durations, potentials of mean force and the directions of net flow in non-equilibrium are shown. In the illustrations of the potentials, the abscissa is the reaction coordinate, and the ordinate represents the potential. These indications are omitted because of space-saving reason. Since the potential of B_1_ < potential of A < potential B_2_, there is a net flow after potential switching that generates an oscillating flow.

**Figure 4 biomolecules-11-00032-f004:**
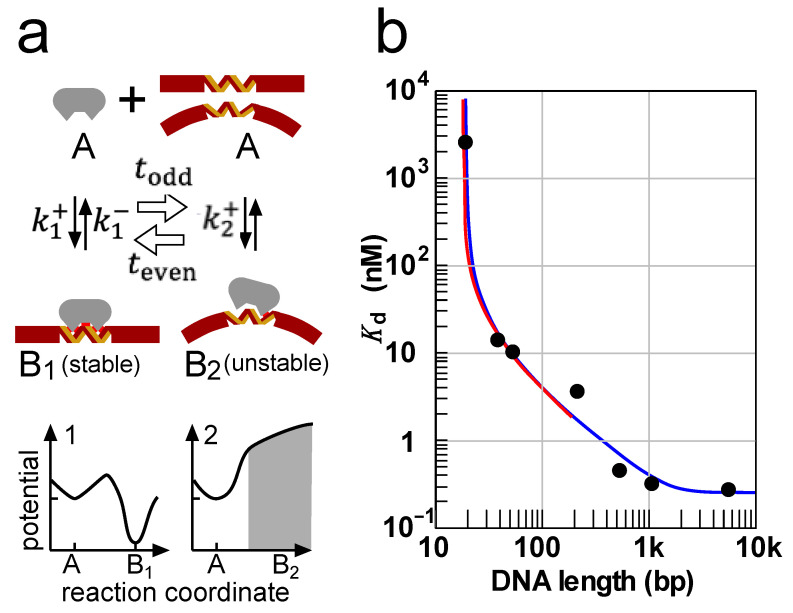
One of the possible molecular models for chemical ratchets in protein–DNA binding. (**a**) Reaction scheme of a chemical ratchet and its alternative Potentials 1 and 2. A DNA binding protein (gray droplets) binds to its specific site (yellow and brown strands) on the DNA (brown bar). The protein stably binds to its specific site on the straight DNA with many cooperative interactions (the four small red boxes in B_1_) in reaction $k_{1}^{\pm }$, and its potential mean force has two local minima. In the presence of one-dimensional diffusion, steps $k_{1}^{+}$ and $k_{1}^{-}$ are both accelerated for longer DNA. In the alternative reaction, $A⇄B_{2}$, B_2_ has only a weak interaction (a small red box) because of a bend at the specific site and is distributed on the high-energy slope (gray) in the potential mean force. Because B_2_ is unstable and because the DNA bent inhibits sliding out of the specific site, B_2_ is destined to be dissociated without molecular shuffling (making it impossible to define $k_{2}^{-}$). Thus, it cannot be converted into B_1_, and B_1_ and B_2_ do not coexist. Since switching from B_1_ to B_2_ rarely occurs due to the enhanced stiffness of the specific site, $t_{odd}$ is larger than the timescale of the binding, making the timescale of the switching slower (larger), even though $t_{even}\;$ may be small. (**b**) The antenna effect of TrpR binding to *trpO*. The values of the dissociation constant (closed circles) were obtained with the site-specific hydroxyl radical footprinting of TrpR on *trpO* DNA of various lengths. The differential equations composed of rate equations and diffusion equations corresponding to the one-dimensional diffusion of TrpR along the DNA were solved in a stationary state to provide the best-fit curve (blue line) with a site size of 18 bp and a diffusion distance of 625 bp (blue) [16]. Another theoretical calculation of chemical ratchet under the assumption of rapid diffusion for short DNA gave the best-fit curve with a site size of 18 bp (red curve) [21], where Panel b is taken from.

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
