# Peer review of "RNA Polymerase and Transcription Mechanisms: The Forefront of Physicochemical Studies of Chemical Reactions"

_biomolecules, 2020, doi:10.3390/biom11010032_

Round 1
Reviewer 1 Report
This manuscript is an introduction to a special issue and a review of the Authors' original point of view on transcription regulation. Their approach is based on a physico-chemical perspective in which protein/DNA interactions take into account molecule diffusion and the antenna effect on long DNA molecules, and the different (active or inactive) complexes formed with the promoter are studied with reaction kinetics taking into additional account the previously introduced concept of chemical ratchet. This review is clearly written and presents the reader with interesting perspectives on a number of parameters that are over-simplified in usual models of molecule interactions in the study of transcription regulation.
Comments
Title
Lines 3, 4 : capital letters missing at "chemical" and "reactions"
Abstract: the Authors should compact their present text that is too general ( the respective contributions of different science domains to the understanding of transcription regulation) in order to give room for a mention of the original points they develop in the manuscript about protein/DNA interactions such as diffusion, reaction kinetics, the concepts of antenna effect and chemical ratchet. It is very important to make the abstract text a good overview of the science presented in the manuscript since readers will decide on its interest upon reading the abstract.
Main text
Line 37: "E. coli the sigma-70 subunit" should be changed to " the E. coli sigma-70 subunit"
Line 62: "This limitation is tried to be overcome..." is very poor English. It could be changed to "Attempts to overcome this limitation with molecular dynamics...
Lines 83, 84: the full equations should be provided there since they are referred to as Eqs.1 and 2 on line 88
Line 98: "non-productive collisions B molecules" should be changed to "non-productive collisions with B molecules"
Line 109: "until the becomes sufficient" a word is missing
Line 110: "the potential barriers>>average energy" please express >>in words
Line 113: "We should aware" should be changed to "We should be aware"
Line 124: "concentration and length are expressed in molar and cm" should be changed to "concentration and length are expressed in molarity and cm"
Line 143:"However but it its difficult" should be changed to "However it its difficult"
Line 180: "based on the overlapping binding site" => binding sites
Line 192: the position of k+ and k- should be aligned with the double arrow <=> on the A + B reaction
Lines 212, 213: please explicit the different promoter fractions. "The promoter is then fractionated into free promoter, inactive promoter complexed with repressor, and active promoter bound by RNAP in open complex"
Line 216: explicit ternary complex: promoter + RNAP + transcript. Or prom+R+I
Line 231: "The scale of the coordinates": please mention the axis
Lines 244,245: this sentence is confusing, please clarify.
Fig3b. – is missing for k1 in the reaction A <=> B1
Fig3c. left part of first figure (under A <=> B1) missing
Line 357: "B2 is not confined by no potential barrier" should be changed to "B2 is not confined by potential barrier"
Line 381, 382: "The longer the DNA is..." the English should be corrected for the whole sentence
Lines 390, 391: please use italics for "in vitro" and for "in vivo"
Line 399: "the diffusion into the promoter by RNAP polymerase" should be changed to "the diffusion of RNA polymerase onto the promoter"
Line 409, 410: please clarify "because of the universal properties"
Line 446: please check the title of reference 18
Author Response
Response to Reviewer 1
This manuscript is an introduction to a special issue and a review of the Authors' original point of view on transcription regulation. Their approach is based on a physico-chemical perspective in which protein/DNA interactions take into account molecule diffusion and the antenna effect on long DNA molecules, and the different (active or inactive) complexes formed with the promoter are studied with reaction kinetics taking into additional account the previously introduced concept of chemical ratchet. This review is clearly written and presents the reader with interesting perspectives on a number of parameters that are over-simplified in usual models of molecule interactions in the study of transcription regulation.
We thank Reviewer 1 for deep understanding. The comments are in blue and the responses are in black. The number in black is the line number in the revised manuscripts.
Title
Lines 3, 4 : capital letters missing at "chemical" and "reactions"
We have corrected it.
Abstract: the Authors should compact their present text that is too general ( the respective contributions of different science domains to the understanding of transcription regulation) in order to give room for a mention of the original points they develop in the manuscript about protein/DNA interactions such as diffusion, reaction kinetics, the concepts of antenna effect and chemical ratchet. It is very important to make the abstract text a good overview of the science presented in the manuscript since readers will decide on its interest upon reading the abstract.
We agree with this comment. The reason why it is too general is that this review is written as the introduction of a special topics of the journal taking account of invited authors. According to the Reviewer 1, when such an introduction is not needed and it can be specific summary of our writing, the abstract is revised as follows. We would like to ask the editor to select either abstract.
Abstract (more specific)
Studies on transcription and its regulation are interdisciplinary and closely connected with genetics, structural biology and reaction theory. Among them, reaction theory has been less stressed but it is getting more critical for further development. To describe reactions involved in transcription, we generally use rate equations, which are based on statistical mechanics and have application limits. We here showed the limits in describing three-dimensional diffusion and one-dimensional diffusion along DNA. Furthermore, we introduce the chemical ratchet mechanism that cannot be described with a single set of rate equations. The mechanism has been proposed for explaining the antenna effect, an enhancement of binding affinity for a specific site on longer DNA that deviates from detailed balance. The ratchet mechanism can be described with alternative sets of rate equations that temporally switch as slow as or slower than the binding reaction.
Main text
We corrected our manuscripts as Reviewer 1 pointed out according to his/her all suggestions. The comments greatly improved the clarity and we thank Reviewer 1’s corrections very much. Revisions are made for the following four comments.
The following 18 comments are accompanied with correct descriptions and we agreed all corrections.
Line 37(37): "E. coli the sigma-70 subunit" should be changed to " the E. coli sigma-70 subunit"
Line 62(62): "This limitation is tried to be overcome..." is very poor English. It could be changed to "Attempts to overcome this limitation with molecular dynamics...
Lines 83, 84(83, 84): the full equations should be provided there since they are referred to as Eqs.1 and 2 on line 88
Line 98(98): "non-productive collisions B molecules" should be changed to "non-productive collisions with B molecules"
Line 110(111): "the potential barriers>>average energy" please express >>in words
Line 113(114): "We should aware" should be changed to "We should be aware"
Line 124(125): "concentration and length are expressed in molar and cm" should be changed to "concentration and length are expressed in molarity and cm"
Line 143(144):"However but it its difficult" should be changed to "However it its difficult"
Line 180(181): "based on the overlapping binding site" => binding sites
Line 192(192): the position of k+ and k- should be aligned with the double arrow <=> on the A + B reaction
Lines 212,213(212,213): please explicit the different promoter fractions. “The promoter is then fractionated into free promoter, inactive promoter complexed with repressor, and active promoter bound by RNAP in open complex
Line 216(217): explicit ternary complex: promoter + RNAP + transcript. Or prom+R+I.
Fig3b. – is missing for k1 in the reaction A <=> B1
Fig3c. left part of first figure (under A <=> B1) missing
Line 357(362): "B2 is not confined by no potential barrier" should be changed to "B2 is not confined by potential barrier"
Line 381, 382(386): "The longer the DNA is..." the English should be corrected for the whole sentence
Lines 390, 391(396,397): please use italics for "in vitro" and for "in vivo"
Line 446(475): please check the title of reference 18
The following five comments are not accompanied by correct descriptions but we have revised these parts according to the comments.
Line 109(109): "until the becomes sufficient" a word is missing.
We revised it as “until the shuffling becomes sufficient”.
Line 231(232): "The scale of the coordinates": please mention the axis
We revise as “The scale of the coordinate, time,”
Lines 244,245(245-7): this sentence is confusing, please clarify.
We revised as “Furthermore, when a corepressor-regulator complex has two or more conformations with different affinities for its operator, this repressor-operator binding can be delayed by the conformational change.”
Line 399(405): "the diffusion into the promoter by RNAP polymerase" should be changed to "the diffusion of RNA polymerase onto the promoter"
We revised as "the diffusion of RNAP onto the promoter"
Line 409, 410(414,415): please clarify "because of the universal properties"
We revised as "forty years ago, an experimental model ~” as "long ago as a coupling between binding and conformational change. An experimental model ~”

Reviewer 2 Report
RNA Polymerase and Transcription Mechanism: The Forefront of Physicochemical Studies of chemical reactions
Nobuo Shimamoto1, 2,*, and Masahiko Imashimizu
The authors studied the direction of information from biology to chemistry and physics in the hope of providing materials or supplies to develop basic science in the other direction across the bridge. The main focus involves transcriptional regulation to achieve their goal.
The manuscript is well written with few errors. The title of the manuscript does not convey the body of the paper correctly. It mentioned RNAP a few times and concentrated more on physicochemical studies of chemical reactions.
Overall, the paper is very theoretical and filled with many assumptions. It is very hard to read the paper to make a good judgement as to the value of the paper to the scientific community. The tone of the paper assumed that many things or past experiments were not right or controversial and their techniques will solve these issues.
Here are several questions or statements I have for the authors.
Lines 50-51: "The actual agreements with these motifs are insufficient". What do you mean by this statement?
The authors should look at two more compilation of E. coli promoter sequences:
Hawleys and McClure, 1983 and Lisser and Margalit, 1993.
A promoter region may contain an UP element, -35 element, extended -10 and -10 element. A perfect -10 and -35 regions cause the RNAP to bind too tightly and decrease the strength of the promoter.
Lines 52-53: "The conservation shows no correlation with promoter strength."
This is not a true statement. There is a correlation of the elements with the strength of the promoters. Changes of the base pair in the -10 or -35 elements can strengthen or weaken the promoter. Changes -11A or -7T will inactivate the promoter.
Lines 53-55: "One of the strongest promoters, the A1 promoter of bacteriophage T7, has a sequence far from the motifs."
This statement is not correct. The T7A1 promoter sequence below has an UP element, a -35 element (5/6 consensus TTGACA), ex-10 (3/4 consensus TATG) and -10 (4/6 consensus TATAAT)
ATTTAAAATTTATCAAAAAGAGTATTGACTTAAAGTCTAACCTATAGGATACT
Line 55: "The deductive prediction of relevant promoters and their strengths remains a challenge." I don’t think the strength of the promoters remains a challenge.
Line 109: Please change "state until the becomes sufficient" to "state until they become sufficient."
Lines 144-145: However, but it is difficult to rationalize that a potential barrier exists at every segment length, because the structural repeat of nonspecific DNA is 1 base pair along DNA axis.
This statement needs clarification, it is not easy to understand.
Line 153: "sequence as long as the protein."
This sentence is incomplete.
Lines 157-158: "In this model, the site size is confused with the distance between contiguous binding sites."
This sentence should be rewritten. It is not clear.
Lines 212-213: Why "inactive repressor complex"?
Are you referring to closed complex since you mentioned open complex?
Lines 244-245: "but the operator must have the repressor affinity for its operator must be dependent of the conformational change."
This sentence can be rewritten, it is not clear.
Lines 359-360: "Irrespective of this difference from what is generally shown in Figure 3c, this example satisfies the requirement for chemical ratchet."
How can irrespective of this difference… satisfies the requirement? This sentence should be clarified.
Line 383: Binding sites (17-21).
The reference 21, Aki and Adhya, 1997, is not correct to be used as an example with two binding sites and DNA looping. GalR binds to two sites 113 bp apart but does not form a DNA loop without a cofactor, HU, which binds in the apex around +6.5 to stabilize the loop.
Lines 390-391: "There is a serious contradiction between the values of the affinities of TrpR for trpO measured in vitro [22] and those estimated from the TrpR protein levels and the numbers of trpO sites in vivo [23]."
Carey noted that there was a difference between her in vitro data and the in vivo data of Gunsalus et al. 1986 and offered the following explanation:
Although the concentration of TrpR in the cell has been determined under conditions of repression and induction (Gunsalus et al. 1986), it does not seem reasonable to try to rationalize those values with the affinities determined here because there is no way to correlate the two sets of conditions.
I don't understand why the authors said it is a "serious contradiction" after reading Carey explanation for the difference between both results.
Line 399: RNAP polymerase. Please change to "RNAP" or "RNA polymerase".
Author Response
Response to Reviewer 2
The comments are in blue and the responses are in black. The number in black is the line number in the revised manuscripts.
The authors studied the direction of information from biology to chemistry and physics in the hope of providing materials or supplies to develop basic science in the other direction across the bridge. The main focus involves transcriptional regulation to achieve their goal.
The manuscript is well written with few errors. The title of the manuscript does not convey the body of the paper correctly. It mentioned RNAP a few times and concentrated more on physicochemical studies of chemical reactions.
Overall, the paper is very theoretical and filled with many assumptions. It is very hard to read the paper to make a good judgement as to the value of the paper to the scientific community. The tone of the paper assumed that many things or past experiments were not right or controversial and their techniques will solve these issues.
We intended to show what are required for further development rather than pointing out old errors. Thus we tried to emphasize the direction in the revised manuscripts as follows.
Line 139
is used for approximation, à contributed to the understanding of the diffusion. Line 396-399
We suggested consistent specificity of TrpR-trpO binding measured in vitro and in vivo, which had been supposed to a discrepancy between the conditions.
Short summary
In the revised manuscript, we separated short summary from Section 8 and added the third suggestion by Reviewer 3 as an example of future prospects. Line 395-399
Here are several questions or statements I have for the authors.
Lines 50-51(52-54): "The actual agreements with these motifs are insufficient". What do you mean by this statement?
We agree with that the sentence lacks clarity. We revised the corresponding sentence as “However, physiologically relevant and strong promoters usually do not have the full consensus motifs.”
The authors should look at two more compilation of E. coli promoter sequences:
Hawleys and McClure, 1983 and Lisser and Margalit, 1993.
We agree with this comment. We cited (Hawleys and McClure, 1983) and (Lisser and Margalit, 1993) in Refs 4 and 5.
A promoter region may contain an UP element, -35 element, extended -10 and -10 element. A perfect -10 and -35 regions cause the RNAP to bind too tightly and decrease the strength of the promoter.
We agree with this comment. We add a sentence on UP element as follows: UP element just upstream of the -35 motif may also be included in the consensus sequences. (Line 50)
Lines 52-53(53,54): "The conservation shows no correlation with promoter strength."
This is not a true statement. There is a correlation of the elements with the strength of the promoters. Changes of the base pair in the -10 or -35 elements can strengthen or weaken the promoter. Changes -11A or -7T will inactivate the promoter.
We agree with this comment and our description was not correct. We thus revised it as follows: except for the best conserved −11A and −7T bases of the −10 motif, the conservation of the entire motif sequences shows no correlation with promoter strength.
Lines 53-55(: "One of the strongest promoters, the A1 promoter of bacteriophage T7, has a sequence far from the motifs."
This statement is not correct. The T7A1 promoter sequence below has an UP element, a -35 element (5/6 consensus TTGACA), ex-10 (3/4 consensus TATG) and -10 (4/6 consensus TATAAT)
ATTTAAAATTTATCAAAAAGAGTATTGACTTAAAGTCTAACCTATAGGATACTWe agree with this comment. We removed the corresponding sentence.
Line 55(56,57): "The deductive prediction of relevant promoters and their strengths remains a challenge." I don’t think the strength of the promoters remains a challenge.
We agree with this comment. We thus revised it as follows: The deductive prediction of relevant promoters and their strengths are discussed in this Special Issue [6].
Line 109(109): Please change "state until the becomes sufficient" to "state until they become sufficient."
We have corrected our mistake.
Lines 144-145(144-145): However, but it is difficult to rationalize that a potential barrier exists at every segment length, because the structural repeat of nonspecific DNA is 1 base pair along DNA axis.
This statement needs clarification, it is not easy to understand.
We agree with this comment. We thus clarified it as follows:
”However, it is difficult to rationalize that a potential barrier exists at every segment length of the protein size (Figure 1a).”
Line 153(152,153): "sequence as long as the protein." This sentence is incomplete.
We thank this comment. We have corrected it as follows: sequence regardless of the protein size
Lines 157-158(157,158): "In this model, the site size is confused with the distance between contiguous binding sites."
This sentence should be rewritten. It is not clear.
We agree with comment. We thus revised it as follows:
Another problem with this model is that the size of the protein-binding site is confused with the distance between two contiguous protein-binding sites (Figure 1a)
Lines 212-213(212,213): Why "inactive repressor complex"?
Are you referring to closed complex since you mentioned open complex?
Yes, this part was confusing and we thus corrected the part as “free promoter, inactive promoter complexd with repressor, and active promoter bound by RNAP in open complex”
Lines 244-245(245-247): "but the operator must have the repressor affinity for its operator must be dependent of the conformational change."
This sentence can be rewritten, it is not clear.
We agree with this comment. We revised that sentence as “Furthermore, when a corepressor-regulator complex has two or more conformations with different affinities for its operator, this repressor-operator binding can be delayed by the conformational change.”
Lines 359-360(363-365): "Irrespective of this difference from what is generally shown in Figure 3c, this example satisfies the requirement for chemical ratchet."
How can irrespective of this difference… satisfies the requirement? This sentence should be clarified.
We clarified the sentence as follows: Although the potential around B2 is not the same as is shown for B2 in Figure 3c, this example also satisfies the requirement for chemical ratchet.
Line 383(388): Binding sites (17-21)
We corrected it. Thank you.
The reference 21, Aki and Adhya, 1997, is not correct to be used as an example with two binding sites and DNA looping. GalR binds to two sites 113 bp apart but does not form a DNA loop without a cofactor, HU, which binds in the apex around +6.5 to stabilize the loop.
We agree with this comment. We removed the reference 21 (Aki and Adhya, 1997) from the citations. Please note that Ref numbers of the revised manuscript are different from those of the original manuscript sent for review.
Lines 390-391(396-399): "There is a serious contradiction between the values of the affinities of TrpR for trpO measured in vitro [22] and those estimated from the TrpR protein levels and the numbers of trpO sites in vivo [23]."
Carey noted that there was a difference between her in vitro data and the in vivo data of Gunsalus et al. 1986 and offered the following explanation:
Although the concentration of TrpR in the cell has been determined under conditions of repression and induction (Gunsalus et al. 1986), it does not seem reasonable to try to rationalize those values with the affinities determined here because there is no way to correlate the two sets of conditions.
I don't understand why the authors said it is a "serious contradiction" after reading Carey explanation for the difference between both results.
We agree with this comment. We thus revised it as follows: There is a discrepancy between the values of the specificities of TrpR- trpO binding in vitro [24] and that expected from the TrpR protein levels as well as the numbers of trpO sites in vivo [25], which was originally reduced to the possible different conditions. Notably, if the antenna effect observed in a similar condition [11] is taken into account, the discrepancy will disappear.
Ref 24. Carey, J. Gel retardation at low pH resolves trp repressor-DNA complexes for quantitative study. Proc. Natl. Acad Sci. USA. 1988, 85, 975-979.
Ref 25. Gunsalus, R. P.; Miguel, A. G.; Gunsalus, G. L. Intracellular trp repressor levels in Escherichia coli. J. Bacteriol. 1986, 167, 272-278.
Line 399(405): RNAP polymerase. Please change to "RNAP" or "RNA polymerase".
We thank this comment. We changed it to “RNAP”.

Reviewer 3 Report
The authors discuss the significance of DNA protein binding and chemical kinetics in the mechanism of gene transcription. The article is interesting and discusses several new and relevant topics such as chemical ratchet as applicable to transcription kinetics. However, I wish the authors did a more thorough revision before sending in the manuscript. There are several places where sentences are unclear and given that some of the concepts are complicated to convey, its important that the text retains utmost clarity.
I think, using the term chemical reaction is a bit misleading, since biochemical mechanisms are intertwined between non-covalent binding, diffusion and enzymatic reactions. The authors discuss all of these important mechanisms within the manuscript. Hence, using the term chemical reaction does not seem to do justice to the topic thats been discussed.
I think, it would be important to delineate the process of transcription and the major steps involved in the process at the beginning in the introduction, before going into detailed discussion.
The role of transcriptional noise due to infrequent transcription is another subject that is linked to the kinetics of the transcription process.
The authors mention degradation as an avenue that depletes mRNA, thereby affecting the timescale of RNA accumulation. There are other processes whose timescales may also be relevant here, such as splicing of immature mRNA immediately after transcription, or the export of mature mRNA to the cytoplasm, which also affect the accumulation of mRNA within the nucleus.
Lastly, the manuscript ends rather abruptly, right after the discussion on chemical ratchets. A short summary section could be helpful here, along with the discussion of outstanding questions that need to be addressed in the future.
Author Response
Response to Referee 3
The comments are in blue and the responses are in black. The number in black is the line number in the revised manuscripts.
The authors discuss the significance of DNA protein binding and chemical kinetics in the mechanism of gene transcription. The article is interesting and discusses several new and relevant topics such as chemical ratchet as applicable to transcription kinetics. However, I wish the authors did a more thorough revision before sending in the manuscript. There are several places where sentences are unclear and given that some of the concepts are complicated to convey, its important that the text retains utmost clarity.
I think, using the term chemical reaction is a bit misleading, since biochemical mechanisms are intertwined between non-covalent binding, diffusion and enzymatic reactions. The authors discuss all of these important mechanisms within the manuscript. Hence, using the term chemical reaction does not seem to do justice to the topic thats been discussed.
One of the author, NS, strongly agrees with Referee 3. The word “chemical ratchet” was cast by Dr. Toda Mikito, a theoretical biophysisist, in ref. 11, where TM and NS are both corresponding authors. I had proposed “kinetic ratchet” instead, because “chemical” seems to suggest covalent reaction. However, Mikito and other two theoretical chemists in the authors were strongly against “chemical~covalent” as too old notion and claimed that chemical reaction involves diffusion and non-covalent binding. Since this article must attract theorists, too, I agreed to select wider and modern definition of chemistry. Anyway, this article is not suitable place for renaming the mechanism.
I think, it would be important to delineate the process of transcription and the major steps involved in the process at the beginning in the introduction, before going into detailed discussion.
Thank you for this comment. We added the simple mechanism of initiation assumed in our qualitative analysis of timescales. Line 205
The role of transcriptional noise due to infrequent transcription is another subject that is linked to the kinetics of the transcription process.
Reviewer 3 is very correct. The mechanism of chemical ratchet is based on the difference in the microscopic behaviors of reactant molecules that disappears if the consideration is limited to the macroscopically averaged behavior of many molecules. We believe that similar discussions are possible in transcriptional noise. We hope several researchers may submit papers in future on the topic. Line 425,426
The authors mention degradation as an avenue that depletes mRNA, thereby affecting the timescale of RNA accumulation. There are other processes whose timescales may also be relevant here, such as splicing of immature mRNA immediately after transcription, or the export of mature mRNA to the cytoplasm, which also affect the accumulation of mRNA within the nucleus.
We agree with this comment. We limited our qualitative analysis of timescales within E. coli transcription and this limitation is described in the revised manuscript. Line 204
Lastly, the manuscript ends rather abruptly, right after the discussion on chemical ratchets. A short summary section could be helpful here, along with the discussion of outstanding questions that need to be addressed in the future.
We separated short summary from Section 9 and added the third suggestion by Reviewer 3

Round 2
Reviewer 2 Report
Second review for Biomolecules
RNA Polymerase and Transcription Mechanism: The Forefront of Physicochemical Studies of Chemical Reactions
The authors studied the direction of information from biology to chemistry and physics in the hope of providing materials or supplies to develop basic science in the other direction across the bridge. The main focus involves transcriptional regulation to achieve their goal.
Thank you to the authors for addressing my questions and for making changes that I suggested.
I am still having a lot of problem with this manuscript. It is well written but sometimes hard to follow. The tone of the paper needs to change with the choices of words. The reader would come away thinking everything which was done in the past was incorrect and this paper would correct those issues. I am a biologist, and I do not understand the chemistry and physics sections.
My recommendation as before to the editor is to distribute the manuscript to a physics expert who will be able to make a better judgement of the merit of the paper because of all the chemistry and physics which are heavily involved in the manuscript. Therefore, I do not feel as if this paper is being judge fairly. It should be submitted to an expert with knowledge of chemical ratchet.
I will reject the manuscript because in this current form of the paper, it is not suitable for publication and to have an impact on the science field.
Here are several questions or statements I have for the authors if they wanted to make any corrections and resubmitted it to the journal or another journal. These are suggestions and the authors should decide whether they want to change them or not. I hope these changes will improve the paper.
The "specific" abstract is better than the "general" abstract.
Line 18: Change "topic" to "paper"
Line 31: delete "described"
Line 33: Change "blind use" to another word. This is not nice to speak of your fellow colleagues.
Line 34: Rewrite "mislead interpretations still surviving"
Line 37: Rewrite "subsequently simplified the biochemical analysis of transcription"
Line 38: Change "The single protein" to "Sigma-70"
Line 44: DNA as its cofactor: DNA is not considered as a cofactor? Cofactors directly influence how RNAP transcribes on the DNA.
Line 48: Change " major sigma factor-----Sigma-70" to "major sigma-70 promoters of E. coli"
Lines 48-49: Change "consist of a TATAAT (-10 motif) and a TTGACA (-35 motif) are ----sequence" to "consist of -10 motif (TATAAT, consensus) and a -35 motif (TTGACA, consensus)"
Lines 50-51: UP element ------- sequences" to "UP element sequence located upstream of the -35 motif may play a role in determining the strength of the promoter".
Lines 54-55: (The conservation of the entire motif sequences shows no correlation with the promoter strength" is not a true statement. Each pair affects promoter strength.
Lines 55-56: The sentence "This lack of correlation----promoters" is also incorrect and weaken the paper.
Line 61: "must be critical" need to explain more and "must" is too strong a word, it can be replaced by "may"
Lines 65-68: These sentences should be rewritten. The information does not flow.
Lines 68-72: These sentences should be rewritten. This diffusion is not clear. Maybe it should read "This diffusion of RNAP complex…
Line 70: Change "Introduced later in the paper." Discussed below."
Line 71: Change "primitive", not a good choice of word.
Line 75: Change "The ignorance of the limit". The word "ignorance" is not appropriate. It is degrading.
Lines 86-90: The choice of words "Not universal true, implicitly, several assumptions, contradicting" should be change. It does affect the interpretation of the manuscript to readers.
Lines 93-94: Define "histories/history"
Line 96: The larger the possibility. Possibility of what?
Lines 102-104: Rewrite, example: To avoid the ambiguity of "temperature", we will use "molecular shuffling" instead of "thermal equilibrium" because thermal equilibrium is used between the two states in a chemical reaction.
Line 111: "barriers are" to "barrier which are"
Line 114: "be aware" to "be awared"
Line 126: "Since --- molecules" is incomplete. Rewrite.
Line 134: Choice of words, "inconsistent, harmless type, over-extrapolation"
Lines 157-176: Choice of words, "another problem, underestimates, model ignores, approximated, mistakenly assumes, model ignores, underestimates" please rewrite. Too many negative words.
Line 205: The words "ternary complex" has a double meaning. Here you mean RNAP, DNA and RNA instead of Repressor, RNAP and DNA.
Line 211: Change "than the following formation" to "than the formation"
Line 217: Change "Ternary complex, promoter+RNAP+ transcript" "Ternary complex, RNAP + DNA + mRNA"
Author Response
Dear Academic Editor
We here respond to the comments by academic editor in blue in confidence.
Editor’s note: As I interpret it, reviewer 2 has chosen to reject primarily as they feel that they cannot make a judgment on the physics aspects of the manuscript. Since the other two reviewers are suggesting to accept the manuscript, I would also suggest that this manuscript can be accepted if the final corrections suggested by each of the reviewers are made.
We responded to Reviewers 1 and 3, who did not respond to our revision in 2nd round, and the response has been already reflected in our previous revision. For the comments by Reviewers 2, we responded to his/her comments as shown below the line “Response to 2nd round comments by Reviewer 2” except illogical ones, as follows.
He/she insists on removing the words “assumptions” and “Not universal truth”. They are cited from Ref. 9, a widely known textbook for the foundation of reaction theory. In addition to his comment requesting “assumption” and “contradicting”, these words are essential to the clarity of a theory. The aim of “Biomolecules” is “to encourage scientists to publish their experimental and theoretical results in as much detail as possible”. In contrast, his comments lack a respect to theory and “contradict”your aim. As you told us, he/she honestly confessed his lack of knowledge to judge our manuscript. Thus he/she should reject reviewing this manuscript but should be neutral on the acceptance of our manuscript.
Reviewer 2’s comment. My recommendation as before to the editor is to distribute the manuscript to a physics expert who will be able to make a better judgement of the merit of the paper because of all the chemistry and physics which are heavily involved in the manuscript. Therefore, I do not feel as if this paper is being judge fairly. It should be submitted to an expert with knowledge of chemical ratchet.
The diffusion-limited association is a summary of a section in NS’s article Cem. Rev. [10], and chemical ratchet is based on in our publication Sci. Rep [16]. Our second article on chemical ratchet (Ref. 21) is going to be accepted for Scientific Reports by one of the editorial boards, Dr. Albert Jeltsch, who has been engaged in the field. The letter by the board says “To ensure the Editor and Reviewers will be able to recommend that your revised manuscript is accepted, please pay careful attention to each of the comments that have been pasted underneath this email. This way we can avoid future rounds of clarifications and revisions, moving swiftly to a decision.”
Editor’s note: Importantly, this manuscript needs to be reviewed by a native or highly-proficient English speaker. Reviewer 2 has done an excellent job identifying several confusing or incorrect sentences, but more thorough editing would greatly benefit both the journal and the authors.
We went through the MDPI English editing before the first revision, but we regret it because it overlooked many parts as you felt on our previous revised manuscript. Thus we went through the edition by “Editage” in this version, and felt a satisfaction.
2nd round comments from Reviewer 2
Comments and Suggestions for Authors
Thank you for the kind and laborious comments. The line numbers are those in the revised manuscript.
I would also recommend using the more specific abstract as requested by reviewer 1.
We switched the abstract that went through the English edition.
Line 18: Change "topic" to "paper"
This word "topic" no longer exists because of the replacement of abstract.
Line 30: delete "described"
Deleted
Line 34: Change "blind use" to another word. This is not nice to speak of your fellow colleagues.
Changed to “careless use”. Thank you for correcting my bad word.
Line 34: Rewrite "mislead interpretations still surviving"
We revised it. It is now logical.
Line 37: Rewrite "subsequently simplified the biochemical analysis of transcription"
The phrase has been deleted.
Line 38: Change "The single protein" to "Sigma-70"
Changed as suggested.
Line 42-44: DNA as its cofactor: DNA is not considered as a cofactor? Cofactors directly influence how RNAP transcribes on the DNA.
Sorry for misleading you. The part has been revised.
Line 48: Change " major sigma factor-----Sigma-70" to "major sigma-70 promoters of E. coli"
Lines 48-49: Change "consist of a TATAAT (-10 motif) and a TTGACA (-35 motif) are ----sequence" to "consist of -10 motif (TATAAT, consensus) and a -35 motif (TTGACA, consensus)"
Lines 50-51: UP element ------- sequences" to "UP element sequence located upstream of the -35 motif may play a role in determining the strength of the promoter".
Lines 54-55: (The conservation of the entire motif sequences shows no correlation with the promoter strength" is not a true statement. Each pair affects promoter strength.
Lines 55-56: The sentence "This lack of correlation----promoters" is also incorrect and weaken the paper.
Line 45-48. Since this introduction is not the place to discuss the problems examined in each article,we deleted all these parts and replaced them with a short and flat introduction of the article in this issue by Imashimizu et al.
Line 52: "must be critical" need to explain more and "must" is too strong a word, it can be replaced by "may"
“May” may be rude to the effort by Chatterji and his colleagues, because it is a logical deduction from genetics by elimination of other possibilities. It is not their speculation. To keep this clear, we revised as “is critical in this case”.
Lines 65-68: These sentences should be rewritten. The information does not flow.
Line 55-58, we revised as
Transcription factors and RNAP, as well as its complexes, are able to diffuse along DNA. For example, this diffusion is involved in promoter search by RNAP, in Brownian ratchet as an elongation complex, and in backtracking of initiation and elongation complexes.
Lines 68-72: These sentences should be rewritten. This diffusion is not clear. Maybe it should read "This diffusion of RNAP complex…
Line 70: Change "Introduced later in the paper." Discussed below."Line 71: Change "primitive", not a good choice of word.
Line 75: Change "The ignorance of the limit". The word "ignorance" is not appropriate. It is degrading.
In Line 61-68, we added more explanation as
As in other biological processes, transcription is composed of chemical reactions. Notably, there are several basic requirements for a chemical reaction to be described with a rate equation [9]. As will be discussed below in detail, chemical processes mediated by transfer via diffusion can be described with rate equations in some cases. However, the transfer itself is beyond the description with rate equations. Furthermore, consideration of the timescales of reactions is critical since timescale matching is essential for the functions of regulators to be expressed, as will be discussed later. Until recently, the mechanism of the chemical ratchet had been overlooked because of the lack of these notions. However, understanding the foundation of scientific tools is essential to promote soundness and clarity in science.
Lines 86-90: The choice of words "Not universal true, implicitly, several assumptions, contradicting" should be change. It does affect the interpretation of the manuscript to readers.
We deleted “implicitly” according to the comment. However, “not universal true” and “assumptions” are exact citation from the reference 7, a widely-used textbook of theory of reaction, and actually we believe that these words are logically exact. Thus we added the citation “[7]” to show no exaggeration by us. Moreover, “assumption” and “contradicting” are correct, sound, and logical words in theory as well as experiment. We here show the remark for rate equations in the textbook “Stochastic Processes in Physics and Chemistry, 3rd Ed” by van Kampen pp 171.
*******************************************************************
Remark. This equation is, of course, not a universal truth, but holds .when the following physical requirements are satisfied.
(i) The mixture must be homogeneous in order that the density at each point of D equals nj/D. For sufficiently slow reactions homogeneity can be achieved or approximated by stirring. Departures from homogeneity are the subject of chap- ter XIV.
(ii) The elastic, non-reactive collisions must be sufficiently frequent to ensure that the Maxwell velocity distribution is maintained. Otherwise the collision frequency could not be proportional to the product of densities, but more details of the velocity distribution would enter. This requirement will be satisfied in the presence of a solvent or an inert gas, but is actually better obeyed than might be expected.
(Hi) The internal degrees of freedom of the molecules are also supposed to be in thermal equilibrium, with the same temperature T as the velocities. Otherwise the fraction of collisions that result in a reaction would depend on the details of the distribution over internal states, and not just on the concentrations. Long-lived excited states, however, may be taken into account by listing them among the Xj as a separate species·), but a clear-cut difference in time scales is indispensable.
(iv) The temperature must be constant in space and time in order that one may treat the reaction rate coefficients as constants even though they depend strongly on temperature.
These assumptions may not be very realistic in many actual chemical reactions, but they do not violate any physical law and their validity can therefore be approximated to any desired accuracy in suitable experiments. They ensure that the state of the mixture is fully described by the set of numbers {nj}'
******************************************
Lines 93-94: Define "histories/history"
Line 85-6: According to this comment, we added
“(where the molecules positioned, how large and in which direction was their momenta at past)”
Line 96: The larger the possibility. Possibility of what?
Line 86-7: We added words as
the shorter the distance between A and B, the larger the possibility for producing C.
Lines 102-104: Rewrite, example: To avoid the ambiguity of "temperature", we will use "molecular shuffling" instead of "thermal equilibrium" because thermal equilibrium is used between the two states in a chemical reaction.
We corrected as suggested, but inserted “also” before “used”.
Line 111: "barriers are" to "barrier which are"
Corrected as “barriers that are.”
Line 114: "be aware" to "be awared"
We deleted this part.
Line 126: "Since --- molecules" is incomplete. Rewrite.
It has been corrected.
Line 134: Choice of words, "inconsistent, harmless type, over-extrapolation"
Line 121-3: we revised as
“It may be possible to use the value of kdiff in Eq. (3) as the upper limit of the rate constant of bimolecular association. However, kdiff cannot be used in a rate equation as a rate constant. “
Lines 157-176: Choice of words, "another problem, underestimates, model ignores, approximated, mistakenly assumes, model ignores, underestimates" please rewrite. Too many negative words.
We used these words because the segment model has many critically negative aspects. We should point them out. But the reviewer is probably afraid of emotional rejection of the manuscript by readers using a segment model. Therefore, we tried to describe the negative aspects with a flat tone.
Line 144-58:
The size of the protein-binding site is different from the distance between contiguous protein-binding sites (Figure 1a and 1b). The distance between the sites is one base pair irrespective of the size of the protein because a protein-binding site is defined at every structural repeat of the polymer (i.e., one base pair). In contrast, the site size is usually equal to the size of the protein on the DNA, typically 15–40 base pairs, unless the protein forms a ring or helical polymer with DNA at its axis, such as LecA [11]. Therefore, at low protein concentrations where the diffusion along the exposed region of DNA takes place without any collisions, there are much more empty protein-binding sites on DNA than the segment model supposes. These binding sites are at least accessible from bulk. When the effect of nonspecific sites is discussed with a segment model, this entropic effect should be taken into account.
The competitive nature of nonspecific binding to DNA is required to evaluate the number of nonspecific complexes. The binding at a site should block additional neighboring sites on both sides. The length of the blocked sites is determined by the protein size as well as the shape of the protein complexed with the helically arranged binding sites on DNA. Since this characteristic mode of competition and the high density of nonspecific sites are based on the structure of double-stranded DNA, it is not easy to incorporate these aspects of one-dimensional diffusion in the segment model.
Line 162-5:
At present, one-dimensional diffusion can be analyzed strictly by a diffusion equation on a continuum DNA only at low protein concentrations where the interference between two protein molecules on the same DNA molecule is ignored. The construction of theoretical framework based on the overlapping binding sites as shown in Figure 1b, has been a significant challenge in biology.
Line 205: The words "ternary complex" has a double meaning. Here you mean RNAP, DNA and RNA instead of repressor, RNAP and DNA.
Line 190: Since we discribed “the competitive binding between a repressor and RNA”, the latter is impossible. But we added “RNAP, DNA and transcript” for clarity.
Line 211: Change "than the following formation" to "than the formation"
Corrected as suggested.
Line 217: Change "Ternary complex, promoter+RNAP+ transcript" "Ternary complex, RNAP + DNA + mRNA"
Since we here discuss on the ternary complex before promoter clearance, we added “initiation ternary complex before promoter clearance”.
